

# An overview of the osseous palmar sesamoid in Anura, with the particular case of some *Rhinella* species

Adriana Manzano[1,2,*] and Virginia Abdala[3,4,*]

[1] Cátedra de Embriología y Anatomía Animal. Facultad de Ciencias y Tecnología, Universidad Autónoma de Entre Ríos, Diamante, Entre Ríos, Argentina
[2] Laboratorio de Herpetología, CICyTTP- Consejo Nacional de Ciencia y Tecnología, Diamante, Entre Ríos, Argentina
[3] Facultad de Cs. Naturales e IML, Universidad Nacional de Tucumán, San Miguel de Tucumán, Tucumán, Argentina
[4] IBN CONICET-UNT, CONICET-UNT, Horco Molle - Yerba Buena, Tucumán, Argentina
[*] These authors contributed equally to this work.

Corresponding author
Adriana Manzano,
manzano.adriana@uader.edu.ar

## ABSTRACT

**Background**. Sesamoids are generally regarded as structures that are not part of the tetrapod body plan. The presence of a palmar sesamoid is assumed to serve as a distribution point for the forces of the flexor digitorum communis muscle to the flexor tendons of the digits, which are embedded in the flexor plate. It has been considered that the palmar sesamoid is present in most anuran groups, and it has been suggested that it acts by inhibiting the closing of the palm, preventing grasping. Typical arboreal anuran groups lack a palmar sesamoid and flexor plate, a pattern shared with other tetrapod groups, which can retain a reduced sesamoid and flexor plate. We focus on the anatomical structure of the *Rhinella* group, which includes species that present an osseous palmar sesamoid and climb bushes or trees to avoid depredation or escape dangerous situations, and can exhibit scansorial and arboreal behaviors. We also add data on the bony sesamoids of 170 anuran species to study the anatomy and evolution of the osseous palmar sesamoid within this amphibian group. Our objective is to bring an overview of the osseous palmar sesamoid in anurans, unveiling the relationship between this element of the manus, its phylogeny, and the anuran habitat use.
**Methods**. Skeletal whole-mount specimens of *Rhinella* were cleared and double-dyed to describe the sesamoid anatomy and related tissues. We review and describe the palmar sesamoid of 170 anuran species from CT images downloaded from Morphosource.org, representing almost all Anuran families. We performed an standard ancestral state reconstruction by optimizing two selected characters (osseous palmar sesamoid presence, distal carpal palmar surface) along with the habitat use of the sampled taxa, using parsimony with Mesquite 3.7.
**Results**. Our primary finding is that sesamoid optimization in the anuran phylogeny revealed that its presence is associated with certain clades and not as widespread as previously anticipated. Additionally, we will also be delving into other important outcomes of our study that are relevant to those working in the field of anuran sesamoids. The osseous palmar sesamoid is present in the clade Bufonidae-Dendrobatidae-Leptodactylidae-Brachicephalidae that we named as PS clade, and also in the archeobatrachian pelobatoid *Leptobranchium*, all strongly terrestrial and burrowing species, though with exceptions. The osseous palmar sesamoid is always

present in Bufonidae, but varies in form and size, depending on the mode that they use their manus, such as in the *Rhinella margaritifera* which has a cylindrical one and also grasping abilities that involve closing the manus. The scattered presence of the bony palmar sesamoid among anuran clades raises the question whether this sesamoid can be present with a different tissular composition in other groups.

## INTRODUCTION

### On sesamoids

Sesamoids have been considered structures that do not belong to the tetrapod body plan (*Le Minor, 1987*; *Giori, Beaupré & Carter, 1993*; *Carter et al., 1998*; *Carter, Mikic & Padian, 1998*; *Sarin et al., 1999*). They are elements that appear separated from the limb's skeleton, usually associated with or embedded in a collagenous structure, such as ligaments or tendons (*Vickaryous & Olson, 2007*; *Regnault et al., 2016*; *Abdala et al., 2019*). These structures present a variety of shapes, are vascularized during ontogeny (*Cake & Read, 1995*; *Lazarte et al., 2022*), but with limited vascularization in adults, depending on their size and where they are located (*Yeung & Garg, 2022*). They are primarily related to the limb joints, including digits' joints (*Chadwick et al., 2014*; *Regnault, Pitsillides & Hutchinson, 2014*; *Abdala et al., 2019*). Sesamoids have been associated with various functions, such as a point to disperse forces, tendon distribution and accommodation, or protection against friction or pressure, to improve the ability of tendons to respond to compressive load, and increase muscles leverage (*Carlsöö, 1982*; *Benjamin & Ralphs, 1998*; *Otero & Hoyos, 2013*; *Zhang et al., 2018*). In many cases, they act as a pulley that alters the lines of the pull of the tendons in which they are embedded, improving muscle contraction and efficiency (*Yammine, 2014*). The association between their presence in hypermobile structures such as digit's joints or even manus, and the mobility of these joints is controversial (*Manzano, Fontanarrosa & Abdala, 2019*). In lizards and frogs, their presence could even prevent some manus or digit movements (*Abdala et al., 2009*; *Abdala et al., 2022*).

The sesamoid tissue composition varies, ranging from fibrocartilage to cartilage and bone (*Tsai & Holliday, 2011*; *Regnault, Hutchinson & Jones, 2017*). Genetic signals induce their origin, followed by epigenetic mechanical signals that drive their later growth, conservation, and tissue composition (*Carter, Mikic & Padian, 1998*; *Carter et al., 1998*; *Doherty et al., 2010*; *Eyal et al., 2015*; *Eyal et al., 2019*). In the case of sesamoid bones, they undergo endochondral ossification during ontogeny, beginning as a cartilaginous structure that ossified subsequently (*Yeung & Garg, 2022*). Many of them as, *e.g.*, the patella in mammals (*Samuels, Regnault & Hutchinson, 2017*), are usually small to medium bones that present a constant distribution in the tetrapod taxa and are considered genetically and phylogenetically inherited (*Vickaryous & Olson, 2007*; *Doherty, 2007*; *Doherty et al., 2010*; *Eyal et al., 2015*; *Samuels, Regnault & Hutchinson, 2017*; *Abdala et al., 2019*).

## The palmar sesamoid

Among tetrapods, the described sesamoids of the manus are the pararadial of Lissamphibia, the metacarpal-phalanx and phalanx-phalanx (proximal and distal respectively) sesamoids of Lissamphibia (*Ponssa, Goldberg & Abdala, 2010*), Sauropsids, and Mammalia, and the palmar sesamoid of Lissamphibia and Sauropsids (*Abdala et al., 2019*). The palmar sesamoid is assumed as a point of distribution of forces from the flexor digitorum communis muscle to the flexor tendons of the digits embedded in the flexor plate (*Haines, 1950*; *Abdala et al., 2009*; *Jerez, Mangione & Abdala, 2010*; *Ponssa, Goldberg & Abdala, 2010*). The flexor plate is composed by the flexor tendon of the flexor digitorum longus muscle, with the palmar sesamoid embedded in this tendon, from which the flexor tendons of each digit arises (*Haines, 1950*; *Moro & Abdala, 2006*; *Abdala et al., 2009*). The palmar sesamoid is present in many anuran groups, and absent in tree frogs, such as Hylidae. The typically arboreal anuran groups lack palmar sesamoid and flexor plate (*Manzano, Abdala & Herrel, 2008*; *Manzano, Fontanarrosa & Abdala, 2019*; *Sustaita et al., 2013*; *De Oliveira-Lagoa et al., 2019*), a pattern shared with other tetrapod groups (*Anolis* lizards, for example), though these arboreal species can retain a reduced sesamoid and flexor plate (*Abdala et al., 2009*; *Abdala et al., 2019*; *Sustaita et al., 2013*). It has been suggested that the presence of the palmar sesamoid could prevent the closing of the palm, and, consequently, the grasping (*Abdala et al., 2009*; *Sustaita et al., 2013*). As far as we know, the only study experimentally addressing this collateral effect of the palmar sesamoid presence is that of *Abdala et al. (2009)*. However, the relationship between this structure, the grasping possibilities of a manus lacking it, and arboreality has been repetitively pointed out (*Moro & Abdala, 2006*; *Manzano, Abdala & Herrel, 2008*; *Abdala et al., 2009*; *Sustaita et al., 2013*; *Manzano et al., 2017*; *Abdala et al., 2019*).

## The palmar sesamoid in anurans

In spite of this intriguing relationship between the palmar sesamoid presence or absence with arboreality, there are no studies that deepen this issue within the anuran's groups.

Some groups of Anura are particularly interesting to study the palmar sesamoid and its related issues pointed out above. Among them, we selected the *Rhinella* group, which includes species that climb bushes or trees to avoid depredation or escape dangerous situations, and can exhibit scansorial and arboreal behaviors, such as the *veraguensis*, *festae*, and *margaritifera* groups (*Chaparro, Pramuk & Gluesenkamp, 2007*; *De Noronha et al., 2013*; *Vassallo et al., 2021*). Generalist anurans, such as *Rhinella arenarum* and *R. marina*, commonly named terrestrial or cane toads, in the face of danger, possess the ability to scale their surroundings, despite lacking any particularly noteworthy morphological adaptations for climbing (*Hudson, Gregory & Shine, 2016*; *Vassallo et al., 2021*). All of them have palmar sesamoids, even the exclusive arboreal species, defying the pervasive relationship between the lack of palmar sesamoid and arboreality. *Rhinella* groups have a variety of behaviors in which they use their manus, from support during landing behavior with limited movements, to those with prehensile abilities during behavior exhibited in climbing (*Vassallo et al., 2021*). Based on this, we select some *Rhinella* species both arboreal and terrestrial, to obtain details on the soft and osseous anatomy. To these samples, we add

data on the bony sesamoids of 170 anuran species to study the anatomy and evolution of the osseous palmar sesamoid within this amphibian group in relation to their habitat use. An examination of the palmar sesamoid in frogs will unveil a panorama of the interplay between this component of the manus, the evolutionary history of the species and their environmental utilization.

## MATERIAL AND METHODS

Thirteen all adults specimens of eight species of *Rhinella* were dissected for this study belonging to institutional herpetological collections from Fundacion Miguel Lillo, CICyTTP-CONICET (DIAM) and L. Ponssa's personal collection (Material S1). Since the material used was already deposited in collections, our study did not raise ethical concerns. Skeletal whole-mounts specimens were cleared and double-dyed with Alcian Blue and Alizarin Red S following *Wassersug*'s (*1976*) protocol. Some of the specimens(*R. arenarum* 8 sp.; *R. dorbigny* 8 sp.; *R. margaritifera* 3 sp.; *R. diptycha* 2 sp.) were not cleared and used for anatomical descriptions of muscles and tendons. Specifically, we considered the variation of the sesamoid shape and its relationship to the distal carpal, the flexor plate, and the related muscles (*Rhinella* species, comparative table in Material S2). These specimens were preserved in 70% ethanol and, at the time of observation, temporarily stained with iodine solution to better contrast the structures. Observations, illustrations, and photographs were made with Nikon SMZ1000 and Zeiss stereo microscopes and a Leica DM light microscope equipped with a digital camera and camera lucida. We observed and dissected both left and right carpus. A movement of the flexor plate, with the sesamoid embedded, was achieved by pulling the flexor digitorum communis muscle and the associated tendons in preserved manus (simulating the mechanism of contraction of the muscle) to corroborate the manus movement possibilities when the palmar sesamoid is present. This movement was compared to that of a preserved manus without sesamoid to control the effect of the osseous sesamoid presence.

We also review and describe the osseous palmar sesamoid of both manus of 170 anuran species from CT images downloaded from the Morphosource, and the University of California, representing almost all Anuran families. Since the CT scan images show only bony sesamoids, and it has been reported that fibrocartilaginous sesamoid can be usual in anurans (*e.g., Abdala, Vera & Ponssa, 2017*) our interpretations on sesamoid absences should be taken with caution. We include a description of the carpus of those specimens exhibiting palmar sesamoid as a way to provide the anatomical context of the palmar sesamoid in anurans. From all examined material, we selected three characters (character matrix in Material S3): (A) Osseous palmar sesamoid: 0 = Absent, 1 = Present; (B) Palmar surface of the distal carpals: 0 = concave, 1 = protruded, 2 = flat; (C) habitat use: 0 = terrestrial, 1 = aquatic, 2 = arboreal. Differences in the detail of the descriptions are due to the differences in the display possibilities of the specimens in the database.

We performed a standard ancestral state reconstruction by optimizing the selected characters. We follow the character changes, *i.e.,* character evolution, through a phylogeny based on *Jetz & Pyron (2018)* and *Feng et al. (2017)* for anuran phylogeny, *Pereyra et al.*

*(2015)* for Bufonidae, and *Wiens et al. (2010)* for Hylidae, integrating both morphosource and dissections data. We used parsimony as our methodological criterion for optimal mapping applied with Mesquite 3.70 (*Maddison & Maddison, 2019*). We consider that parsimony is an appropriate methodological option to deal with morphological characters (*Goloboff, 2022*), especially in this context of mapping characters on phylogenies, as the branch lengths are not considered (*Pol, 2001*). Through character mapping, we investigate the relationship between the palmar sesamoid, the carpal ventral surface, phylogeny, and the habitat use within the anuran groups.

## RESULTS

Anatomical descriptions of the bony palmar sesamoid, the associated muscles, and tendinous system showing the variations between four *Rhinella* species with different habitat use were done (Material S2; Figs. 1A, 1B, 1C, 1D, 1E and 1F). The data from the 170 specimens obtained from Morphsource are also recorded. The descriptions are detailed in Material S2 in a comparative manner. Below are the descriptions of the osseous palmar sesamoid and the distal carpal surface anatomy based on specimens' CT images of Morphosource. Only those taxa with osseous palmar sesamoid are included in this list, from which characters A and B were surveyed. Character A, the presence/absence of the bony sesamoid, represents the variation found in the form and distribution of this sesamoid (Fig. 2). Character B, represents the shape of the distal carpals. These carpals are bones that could be independent at the base of each digit (2, 3, 4, and 5) or fused during the development of the anuran carpus as one bone, distal carpal 3+4, or 3+4+5, or 4+5 (*Fabrezi & Alberch, 1996*). The palmar surface of the distal carpal unique bone could be protruded, flat or concave due to two parallel crests located on the edges of the bone. Both characters were used to perform the ancestral state reconstruction of the palmar sesamoid presence and the distal carpal palmar surface.

### Morphosource CT images (Figs. 2A, 2B, 2C, 2D, 2E)

Morphological descriptions of the ventral surface of the manus of species downloaded from Morphosource CT imagens were organized in Material S4. We selected only those species that have sesamoid to describe the shape and the anatomy of the distal carpals that housed it. There is a considerable variation of the width and depth of the concavity, formed by two parallel crests. Those crests could vary in height, if they are present. Some other details interesting to mention were included, such as the hole that faces the base of metacarpal III, a possible pass of nerves and capillary vessels.

### Distal carpals—Sesamoid relationship (Figs. 2A, 2B, 2C, 2D, 2E; 3A, 3B, 3C)

The distal carpal 3+4+5 presents two parallel processes projected from the palmar surface. One is at the preaxial edge of the carpal (as in the case of that of hamalus in the hamate bone of the human manus), and the other on the postaxial edge of the distal carpal. These processes are not always evident, such as in *Hadromophryne natalensis* (Fig. 2D). More developed processes are seen in, *e.g.*, *Rhinella margaritifera* (Fig. 1F), *Truebella tothastes*,

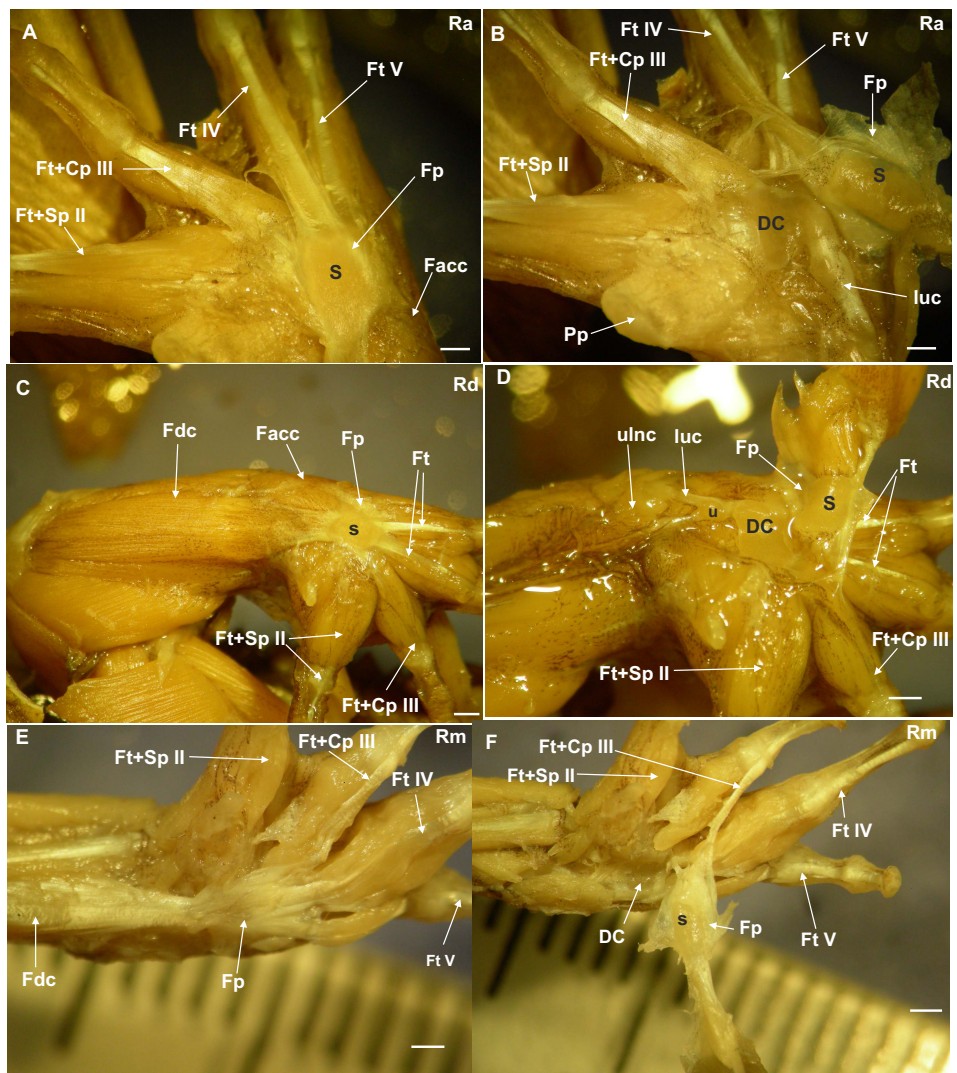

**Figure 1 Ventral surface of the manus with the skin removed.** (A) *Rhinella arenarum* (Ra), ventral view of the superficial musculature of the manus, showing the sesamoid and the flexor plate. (B) *Rhinella arenarum* (Ra), ventral view of the manus. The flexor plate and the sesamoid were raised to show the sesamoid form and the distal carpal 3+4+5 form. (C) *Rhinella dorbigny* (Rd), ventral view of the superficial musculature of the manus, showing the sesamoid and the flexor plate. (D) *Rhinella dorbigny* (Rd), ventral view of the manus. The flexor plate and the sesamoid were raised to show the sesamoid form and the distal carpal 3–5 form. (E) *Rhinella margaritifera* (Rm), ventral view of the superficial musculature of the manus, showing the sesamoid and the flexor plate. (F) *Rhinella margaritifera* (Rm), ventral view of the manus. The flexor plate and the sesamoid were raised to show the sesamoid and the distal carpal 3+4+5 form. DC distal carpal 3+4+5; Facc, Flexor accesorius; Fcr, Flexor carpi radialis; Fcu, Flexor carpi ulnaris; Fp, flexor plate; Ft+Cp III, Flexor tendon digit III+Caput profundum digiti III; Ft+Sp II, Flexor tendon II+superfcialis proprius digit II; Ft IV, flexor tendon digit IV; Ft V, flexor tendon digit V; luc, ulnocarpalis ligament; Pp, prepollex; S, sesamoid; U, ulna. Scale: one mm.

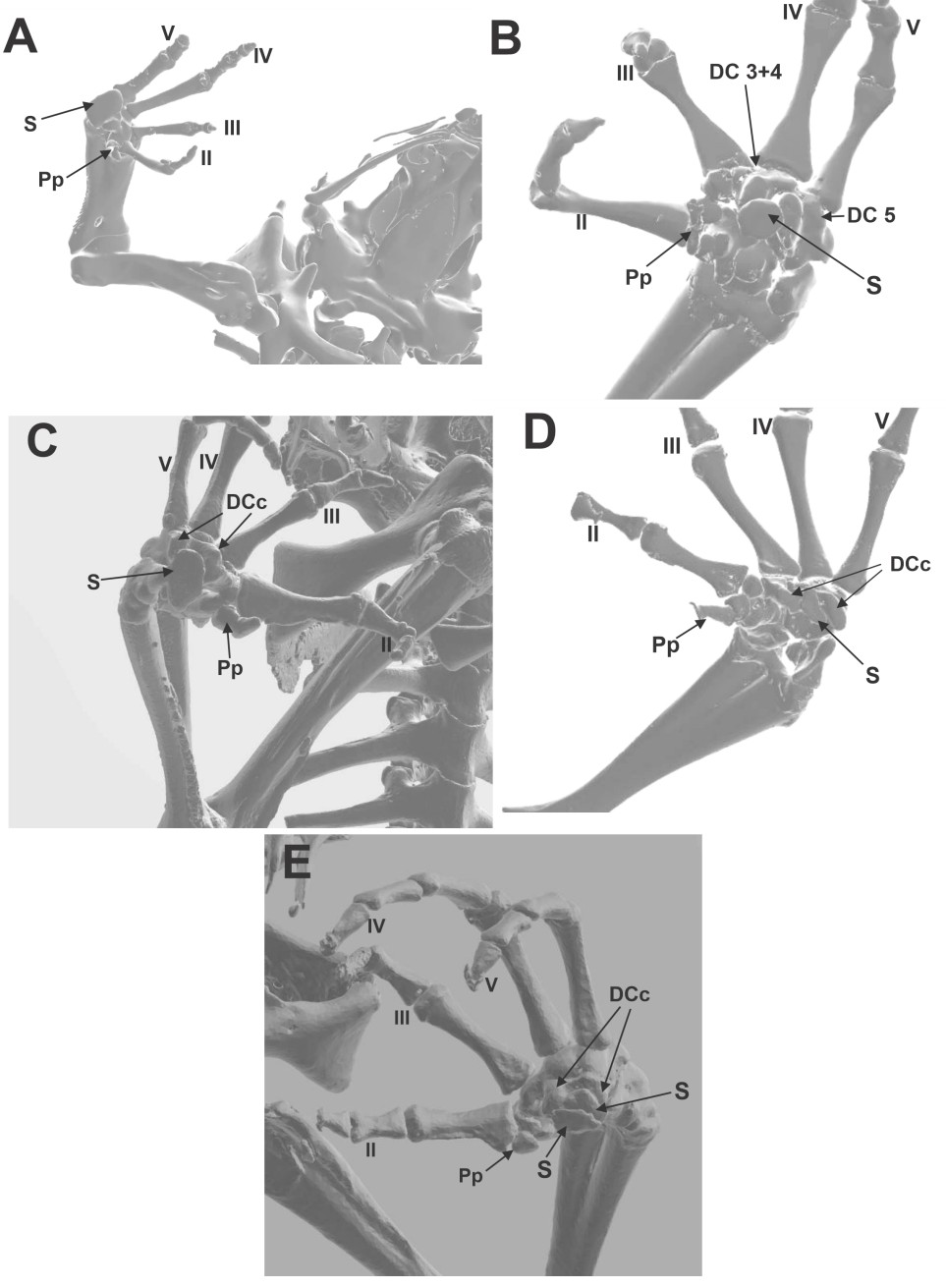

**Figure 2  CT images of the ventral surface of the manus, downloaded from Morphosource.org, University of California, showing the bony palmar sesamoid and the distal carpal crests.** (A) *Hemisus guineensis*. (B) *Leptobrachium haseltti*. (C) *Anaxyrus terrestris*. (D) *Hadrophryne natalensis*. (E) *Melanophryniscus stelzneri*. DCc distal carpal 3+4+5 parallel crest; DC 3+4 distal carpals 3+4; DC 5 distal carpal 5; S sesamoid; Pp prepollex. Digits are enunciated in Roman notation as II, III, IV, and V.

*Anaxyrus terrestris* (Fig. 2C), showing a deep depression between them, which form a concavity. In all cases, the distal carpal is located beside the base of the digits III to V, dorsal to the palmar sesamoid, and the flexor plate. In most cases, the sesamoid embedded on

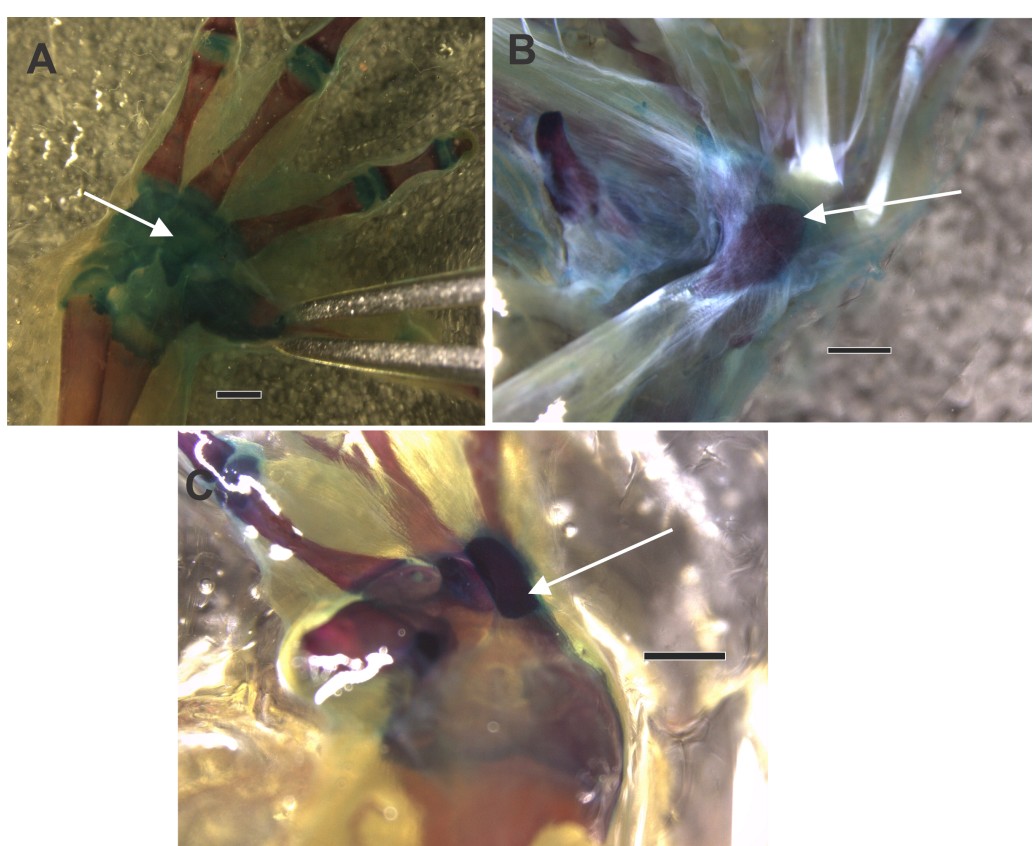

**Figure 3** **The ventral surface of a diaphanized manus shows the osseous palmar sesamoid embedded in the flexor tendon of the m. flexor digitorum longus forming the flexor plate.** Differences in the form of the palmar sesamoid can be seen, with a drop-shaped sesamoid in (A) and (B), and a rectangular sesamoid in (C). (A) *Rhinella arenarum*. (B) *Rhinella henselli*. (C) *Rhinella major*. The black arrow indicates the palmar sesamoid. Scale: 1 mm.

the flexor plate, rests onto the distal carpal, see, *e.g.*, *Hemisus guineensis* (Fig. 2A), or as in the case of *Melanoprhyniscus stelzneri* (Fig. 2E). In *Rhinella*, the development in the size of the palmar sesamoid varies from a small oblong bone to big oval with a proximal process (Figs. 3A, 3B and 3C). In *R. margaritifera*, the sesamoid is cylindrical (Fig. 1F). It fits into the deep depression, among the two parallel processes of the distal carpal, totally covered by the flexor plate and fused to it ventrally. The tendon that integrates the flexor plate varies in size even overpassing the size of the sesamoid.

### Ancestral state reconstruction of the palmar sesamoid presence, morphology of the distal carpal palmar surface, and the habitat use (Figs. 4A, 4B, 4C)

The optimization of the bony palmar sesamoid in the anuran phylogeny (*Feng et al., 2017*; *Jetz & Pyron, 2018*) shows that its absence is a widespread character among anurans, being present in specific, mostly neobatrachia clades; we name this big clade as PS (with bony palmar sesamoid). Thus a phylogenetic association is shown by our data. Interestingly,

the taxa within PS clade are strongly terrestrial, and those secondarily arboreal, such as *Rhinella margaritifera* and *Frostrius*, exhibit a modified shape of the palmar sesamoid. This clade presents some reversions to the ancestral state without palmar sesamoid within the mostly arboreal Hyloydea clade in the dendrobatid *Minyobates minutus*, in the odontophrynids *Proceratophrys bioei* and *Macrogeniaglottus alipioi*; in the hylodid *Crossodactylus gaudichaudii* and *C. trachystomus*. Other taxa exhibiting reversions are the cycloramphids *Zachaenus parvulus* and *Cycloramphus asper;* the miobatrachid *Crinia signifera*, and *Hadromophryne*, a Heleophrynidae, the sister clade of Hyloidea+Ranoidea. Among ranoids, the palmar sesamoid is exhibited in the hemisotid *Hemisus guineensis*; the microhylids *Chiasmocleis crucis*, *Gastrophryne carolinensis*, *Syncope antenori*, and *Melanobatrachus indicus*; the pyxicephalids *Cacosternum boettgeri* and *C. namaquense*. The only taxon with a palmar sesamoid among Archeobatrachia belongs to the Pelobatoidea clade, *Leptobrachium hasseltii*.

The shape of the distal carpal ventral surface shows no clear relationship with the presence/absence of the osseous palmar sesamoid. The presence of the parallel crests on the distal carpal, as edges of a concave surface, is independent of the palmar sesamoid presence (CT image of the skeleton of the ventral surface of *Lymnodinastes dorsalis* can be seen in Material S5).

# DISCUSSION

In this study we present a survey of the osseous palmar sesamoid and its relationships with the carpal surface within the anuran amphibian group, to test whether an association can be proposed between these anatomical structures, the phylogeny of the anuran groups, and the habitat use of these taxa by using an ancestral state reconstruction. We consider that if we find that the selected characters change rapidly in response to varying environments, *i.e.,* habitat use, they should not be well correlated with phylogeny.

The osseous palmar sesamoid optimization in the anuran phylogeny showed that its presence is strongly related to specific clades and not as widespread as expected (*Ponssa, Goldberg & Abdala, 2010*). We can propose that the osseous palmar sesamoid arises early in Neobatrachian groups, being a parallelism or an independent acquisition in the Pelobatoidea *Leptobranchium hasseltii*. Our data indicate that the osseous palmar sesamoid is highly variable in shape and size, with some taxa such as *Melanophryniscus stelzneri*, presenting two overlapping palmar sesamoids (Fig. 4). *Deforel et al. (2021)* described a single, large, and ossified palmar sesamoid as present, in general, in the genus *Melanophryniscus*. However, *Abdala et al. (2022)* did not find this sesamoid in this *Melanophryniscus* species, suggesting that this variability should be further addressed.

The scattered presence of the osseous palmar sesamoid among anuran clades suggests that its genesis can be explained not only by phylogenetic inheritance but also by other factors (such as environmental ones), maybe related to the anuran habitat use. Interestingly, the terrestrial habitat use optimizes as ancestral to the anuran group, which allows us to suppose that the osseous palmar sesamoid absence may indicate a palmar sesamoid of another tissue composition. Bufonidae-Dendrobatidae-Leptodactylidae-Brachicephalidae

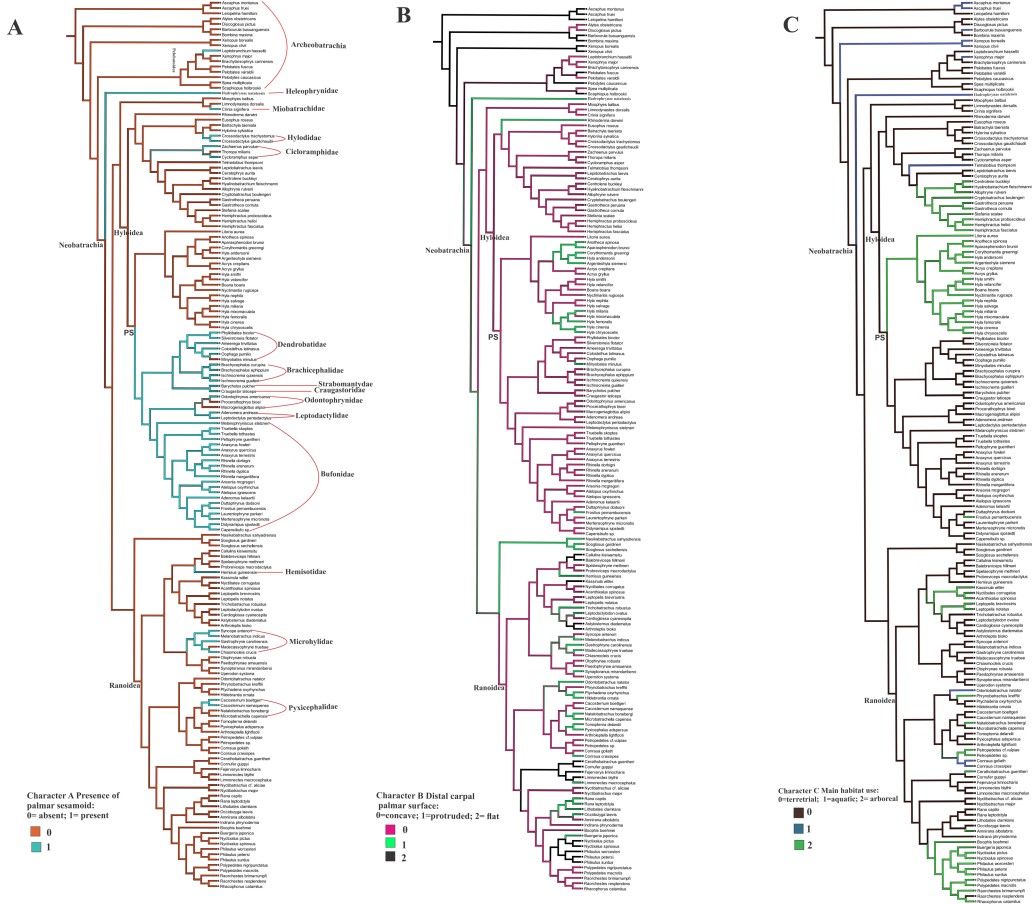

**Figure 4  Ancestral state reconstruction of the osseous palmar sesamoid presence, and the distal carpal palmar surface, taking into account 170 anuran species.** The optimization of characters A and B was based on the anuran phylogeny (*Feng et al., 2017*; *Jetz & Pyron, 2018*). (A) Character A presence of the palmar sesamoid: 0 = absent; 1 = present. (B) Character B distal carpal palmar surface: 0 = concave; 1 = protruded; 2 = flat. (C) Character C Main habitat use: 0 = terrestrial; 1 = aquatic; 2 = arboreal. PS clade with palmar sesamoid.

(PS clade), clearly united by a common ancestor, present a bony palmar sesamoid and share a terrestrial habitat use, inhabiting the forest floor litter and lowland rainforests. *Leptobrachium hasseltii*, an archeobatrachian pelobatoid frog, also presents a bony palmar sesamoid, shows cryptic habits and also similar habitat use, does not belong to this clade, leaving some room for the discussion of non-phylogenetic factors. Tadpoles of these taxa usually live in quiet pools and ponds. *Leptobrachium hasseltii* would represent the ancestral mode of life of frogs (*Herrel et al., 2016*), and it is particularly special since its osseous palmar sesamoid is located between two crests formed by the protrusion edge of the distal carpals 5 and 3 +4 (Fig. 2B). The re-examination of the ancestral state unveils another captivating pattern. It seems that the absence of a crest results in a flat and smooth carpal surface without a palmar sesamoid. Nonetheless, the existence of crests does not hold a direct association with the existence or non-existence of a palmar sesamoid. This suggests

that a different element, apart from the crests, is imperative for the connection of the palmar sesamoid to the carpus.

Among bufonids, almost all species examined have osseous palmar sesamoid and primary terrestrial habitat use, which would be the ancestral state according to *Herrel et al. (2016)*. Functionally, the presence of such a structure in the middle of the manus could prevent the closure of the manus by limiting the digits from touching each other (*Sustaita et al., 2013*). *Manzano, Abdala & Herrel (2008)* and *Manzano, Fontanarrosa & Abdala (2019)* noted that tree frogs could grasp because they not only do not have a bony sesamoid embedded in the flexor plate, but also a reduced or absent flexor plate. Our results evidence the absence of the osseous palmar sesamoid in tree frogs (*Manzano, Abdala & Herrel, 2008*). These data are in accordance with studies in other groups of tetrapods such as lizards and marsupials (*Moro & Abdala, 2006*; *Abdala et al., 2009*; *Abdala et al., 2009*; *Fontanarrosa & Abdala, 2014*; *Fontanarrosa & Abdala, 2016*) supporting the pervasive link between the invasion of the narrow branches niche and the lack or reduction of the palmar sesamoid. The examination and comparison of dissected specimens revealed that a manus devoid of an ossified sesamoid exhibits an increased range of motion. This implies that, from a functional standpoint, the presence of this structure may hinder climbing capabilities. This is a fascinating conclusion, and we are inclined to accept it. However, exceptions do exist, such as in certain species of Bufonidae, such as *Rhinella margaritifera* and *R. paraguas*, which possess an osseous palmar sesamoid and exhibit scansorial behavior, enabling them to ascend shrubs and small trees (*De Noronha et al., 2013*; *Vassallo et al., 2021*; *Ceballos-Castro, Cabra-García & Ospina-Sarria, 2023*). Our data indicate that in such a case, the shape of the sesamoid is a crucial factor, functionally enabling the manus to grasp. The *Rhinella margaritifera*'s palmar sesamoid has a convex surface that articulates with the deep concave depression of the carpal distal 3+4+5, like a hinge joint. This kind of articulation enables the movement of closing the manus around a thin object, such as a small branch, and the sesamoid, in this case, has a cylindrical shape. Curiously, the Bufonidae osseous palmar sesamoid, which can be big or small, has a particular drop shape, and lays superficial to the distal carpals, being usually flat or with a slight depression limited by two crests (Figs. 1 and 3). Our ancestral state reconstruction revealed that the evolution of the distal carpal depression and the sesamoid seem to be independent of each other, suggesting that their relationship is limited to a functional interaction in the manus of *R. margaritifera*. Thus, this association between the palmar sesamoid shape and the carpal surface can be proposed as an exaptation that led to a grasping manus.

*Fontanarrosa, Fratani & Vera (2020)* assigned the palmar sesamoid to the pectoral—forelimb module, formed by the coracoid elements and the forelimb, showing the high intermediacy of this sesamoid, comparable to that of the canonical bones. This high value of this network parameter suggests a functional significance, probably related to the requirements of the strongly terrestrial and burrowing mode of life. Additionally, the pectoral and forelimb region has the important function of absorbing the stress of the impact in the landing phase of the jump (*Emerson, 1982*; *Fontanarrosa, Fratani & Vera, 2020*). It has been proposed that in most frogs, the palmar sesamoid could be participating of the landing mechanism (*Abdala et al., 2022*) by regulating the curvature of the distal

phalanges of the manus. Thus, the palmar sesamoid could be contributing to an indirect mechanical stress absorption during landing.

## CONCLUSION

The optimization of sesamoids in the anuran phylogeny revealed that an osseous palmar sesamoid is strongly associated with specific clades, contrary to what was previously thought. The presence of a bony palmar sesamoid can be observed in the clade made up of Bufonidae, Dendrobatidae, Leptodactylidae, and Brachicephalidae, which we have named the PS clade, as well as in the archeobatrachian pelobatoid *Leptobranchium*. These species are all strongly terrestrial and burrowing, though there are exceptions. Bufonidae always has a bony palmar sesamoid, but its form and size vary depending on how the species uses its manus, such as in *Rhinella margaritifera*, which has a cylindrical one and the ability to grasp through closure of its manus. The scattered presence of the bony palmar sesamoid among different anuran clades raises the question of whether this sesamoid can exist with different tissue compositions in other groups.

## ACKNOWLEDGEMENTS

We extend our gratitude to Morphosource.org and particularly to David Blackburn and Mackenzie Shepard, who graciously permitted us to utilize 3D images and assigned the DOIs to each image. This project was accomplished during the trying times of the pandemic lockdown, and we are immensely thankful to those who advocate for open science, like Morphosource.org. Our appreciation also extends to Santiago Ron and the QCAZ Herpetological Collection for generously donating specimens of *Rhinella margaritifera* to the CICyTTP-CONICET Herpetological Collection and to Laura Ponssa for lending us specimens from her personal collection.

### Funding

This work was supported by the Fondo Nacional de Ciencia y Tecnología de Argentina by PICT 2016-2772, PICT 2018 382; the Consejo Nacional de Ciencia y Tecnología PIP 0389. The funders had no role in study design, data collection and analysis, decision to publish, or preparation of the manuscript.

### Grant Disclosures

The following grant information was disclosed by the authors:
Fondo Nacional de Ciencia y Tecnología de Argentina: PICT 2016-2772, PICT 2018 382.
Consejo Nacional de Ciencia y Tecnología: PIP 0389.

### Competing Interests

Virginia Abdala is an Academic Editor for PeerJ.

## Author Contributions

- Adriana Manzano conceived and designed the experiments, performed the experiments, analyzed the data, prepared figures and/or tables, authored or reviewed drafts of the article, dissections and imagens processing, and approved the final draft.
- Virginia Abdala conceived and designed the experiments, performed the experiments, analyzed the data, authored or reviewed drafts of the article, dissections, and approved the final draft.

## Animal Ethics

The following information was supplied relating to ethical approvals (*i.e.*, approving body and any reference numbers):

Specimens belong to IBN-CONICET-UNT and CICyTTP-CONICET Herpetological Collections.

## Data Availability

The raw data is available in the Supplemental Files.

Dissection Material: *Rhinella arenarum* FML 29863; FML 29859; DIAM 177, 178; L 935; LS 337B; Laura Ponssa personal collection (MLP s/d, 2 specimens)

Rhinella henselii CFBH 20277

Rhinella major FML 29837

Rhinella dorbigni DIAM 344

Rhinella disptyca DIAM 460

Rhinella margaritifera DIAM 514, DIAM 516, DIAM 517

The 219 scans are available at Morphosource (File S1).

## Supplemental Information

Supplemental information for this article can be found online at http://dx.doi.org/10.7717/peerj.15063#supplemental-information.

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
