# Peer review of "An overview of the osseous palmar sesamoid in Anura, with the particular case of some Rhinella species"

_PeerJ, doi:10.7717/peerj.15063_

## Round 0.1 · original submission · Major Revisions

The reviewers broadly have similar comments, which I agree on, too. In particular, a phylogenetic analysis is needed to test for lifestyle-palmar carpal correlation, and some acknowledgement made of correlation/causation issues; and mineralization/ossification/not. But everyone agrees that this is a worthy and interesting paper, so we look forward to the revised manuscript, thank you!

·

Basic reporting

Within the article, the language used is sometimes ambiguous or would benefit from elaboration or clearer wording. I have noted instances that stand out to me in the ‘Additional Comments’.
Appropriate literature and references are cited, but sometimes the citations could be discussed a bit further to given the reader more context, which would be helpful in understanding the study and authors’ interpretations. I have noted instances in the ‘Additional Comments’ where more information and context would be useful to the reader, in my opinion.
The article is structured professionally under appropriate headings and subheadings. However, within sections (e.g. introduction, results, discussion) the narrative or logical flow of content was not always clear to me as a reader. I have noted some specifics/detail in the ‘Additional Comments’, however as a general recommendation I feel re-arranging the content/narrative of the Introduction and Discussion (in particular) into ‘subject paragraphs’ with clearer-signposted ‘topic sentences’ would greatly assist the reader in understanding the study’s rationale, findings and interpretation.
I felt the tables and figures could be improved, or made better use of. I have noted this in more detail in the ‘Additional Comments’. The raw data have not been shared, and the data presented in the some results only include those species for which evidence of a palmar sesamoid was found. In my opinion, for these species, too much descriptive anatomical data are presented – I am not sure of the value of this reporting. It would be better to present the full dataset, with anatomical findings summarized in a way that is more applicable to the questions asked by the study.
The study puts forward some interesting questions, but does not go so far as to formulate these into a hypothesis. Because of that, and the way the results are analysed, it is difficult to interpret the results presented. The authors suggest some exciting functional interpretations based on the study observations, but I feel there are other interpretations that would be equally valid and which are not considered (I have noted these in the ‘Additional Comments’). In its current form, I feel we are only seeing half of a (potentially very interesting) story.

Experimental design

The study is clearly within the aims and scope of the journal, and I believe would be of interest to the journal’s readership (I was very interested to read it!).
As mentioned above (in Basic Reporting), I thought re-organisation and re-writing some parts of the manuscript’s introduction (with subject paragraphs and topic sentences) would assist the authors’ in presenting a well-defined, relevant research question – currently, I feel the narrative and rationale of the study design is somewhat lost. It’s not that the paragraphs or sentences are badly written, but their significance was not always clear to me as the reader – sometimes coming across as a series of statements and references, but not a clear point.
The research does seek to fill an intriguing knowledge gap in understanding why or why not sesamoids might be present, and inferring that for the palmar sesamoids in Anura, it might be to do with terrestrial lifestyle. The investigation and observations are consistent with the interpretations made by the authors; however, there are other interpretations which could also be consistent with these observations and which I do not feel have been rigorously explored. For example, the correlation between palmar sesamoid presence in clades and terrestrial lifestyle has been interpreted to mean that the sesamoid is functionally important to that locomotory mode/lifestyle. However, the lineages (clades) that possess the palmar sesamoid are not independent – could the fact that these closely related lineages both possess the palmar sesamoid and be terrestrial simply be due to shared ancestry, and the one not have any functional bearing on the other? This is not mentioned at all in the study. Further exploration of this possibility, and perhaps implementation of phylogenetic comparative methods, would be a way forward to more rigorously analyse the data.
In order for the study to be replicable, the full list of examined specimens from Morphosource should be detailed and reported (not just those in which there is evidence of the ossified palmar sesamoid).

Validity of the findings

Underlying data have not been fully provided.
I have concerns that the main conclusion is not well-supported by the results. The research question and main conclusion of the study seems to be that palmar sesamoid presence in Anura is related to terrestrial locomotion/lifestyle, but this is not tested or illustrated by the data (e.g. coding locomotory mode on the phylogeny to compare with sesamoid presence). A few clades or species are picked out to illustrate the apparent correlation between lifestyle and sesamoid presence, but it does not appear that the lifestyle of the other clades has been systematically described or plotted/reconstructed over the phylogeny to support that interpretation. There are also other interpretations of the study observations that don’t appear to have been explored (e.g. phylogenetic non-independence, correlation vs causation between lifestyle and sesamoid presence).
A more minor comment is that sesamoids may be ossified, mineralized, or cartilaginous. The CT modality through which most of the specimens were assigned sesamoid status in this study is often unable to discern cartilage tissues as well as bone. Therefore, interpreting sesamoids as ‘absent’ in certain lineages for the purposes of phylogenetic reconstruction may be misleading. The study question and discussion may need to be re-centred / re-worded to more explicitly focus on mineralized/ossified sesamoids.

Additional comments

Overall comment – This interesting study seeks to address a real knowledge gap in sesamoid presence and distribution, with some intriguing correlations apparent. I also commend the authors for continuing this important avenue of inquiry during a difficult period (lockdown). However, I have concerns that only one interpretation is really offered in this study, with no real exploration or interrogation of the data for other, perhaps equally valid, explanations for the correlations that seem to be observed. There are also improvements that could be made in terms of structure and narrative, and the full data need to be provided or at least described.

I apologise to the authors - I was unable to annotate the PDF directly, so I have made fuller Additional Comments referencing the PDF to be attached.

·

Basic reporting

Good, a table could help (see section 4)

Experimental design

See above

Validity of the findings

Seem valid, some phylogenetic regressions could help

Additional comments

Here, I have the pleasure of reviewing An overview of the palmar sesamoid in Anura, with the particular case of some Rhinella species by Manzano and Abdala. I must first and foremost apologies for not spending the time on this manuscript that it deserves. I accepted this review because I am a fan of the work from this lab, and was looking forward to reading their latest research. Unfortunately, it is one of those times in life when, to do what I need to, I am sitting at a computer at 8:30 PM (after having started working at 7:30 AM) just to hit submit on this review, two weeks after I was originally supposed to submit it. Because of time constraints, this review is shorter and less in depth than it should be, and I apologize for that.

I very much enjoyed reading this manuscript. As the authors already know, it is quite anatomically heavy. I was wondering if it would be possible to have the results from L129-284 presented in a table, instead of as paragraphs of written text (or both)? The table could have information of taxonomy, specimen number, and then summarized details from the paragraphs. This might make the information easier to digest, compare within and across taxa, and it might make the information more user-friendly for future researchers who include this data in their own work. Included in this table, a column of “method of data collection” would be useful as well (i.e., dissection vs microCT).
Another small thing – if available, could a figure be created showing all different types of palmar sesamoids and carpal depressions side-by-side, so the readers have a fuller understanding of differences in morphology?
I was also wondering if the authors would be willing to take the phylogenetic analyses a step further. They qualitatively discuss the lack of a relationship between the distal carpal depression and the sesamoid, but do not test this relationship. This could be done with a phylogenetic regression and would aid the paper. Similarly, a regression could investigate the relationship between mode of locomotion and these morphological characteristics. It would help bolster their conclusions about the relationship between sesamoids and locomotion.

Some smaller comments are below.
L32: avoiding -> to avoid
Introduction
The flexor plate is mentioned a few times in the Abstract/Introduction – it would be useful to have defined and/or shown on an anatomical image – you can just refer to Table 1 and Figure 1 if you would like, but – as someone who doesn’t study non-mammalian animals – I am unfamiliar with this structure and have trouble identifying it and, in particular, differentiating it from the palmar sesamoid in the images in Figure 1.
L187: lyings -> lying
L220-260 (and potentially elsewhere): the word distal should not be capitalized
L233-4: “Distal carpal 3-5 with a big protuberance.” Not sure what is meant by this sentence
L242: unsure if 3 + 5 is supposed to be 3 – 5
L322: digit’s not digits

Figures
Fig 1 caption – spelling mistake in “fexor plate” – please check other anatomical names
Fig 4 caption – ûat Should be flat I think

---

## Round 0.2 · Minor Revisions

We apologise for the delay in finalising this round of reviews. Northern Hemisphere winter holiday time and other factors contributed. The reviews give some tips on how to improve the wording, and some more proofreading would be helpful.

Reviewer 2 has some more substantive critiques-- I'll attempt to intermediate:
on optimisation, there are multiple meanings of this word in phylogenetic comparative methods. Optimisation of characters, whether on trees or to build the trees themselves, has long been used; this meaning of optimisation can be but is not necessarily identical to parsimony (one could say that parsimony is a method used in optimisation). There are way too many references as examples but one arbitrarily is:
https://bmcbioinformatics.biomedcentral.com/articles/10.1186/s12859-015-0675-0
(about building trees)

Anyway, if the revision could make it extra clear that the method is standard character optimisation (mapping, tracing, ancestral state reconstruction, etc.), that would help. I think performance optimisation (i.e. maximal jumping ability, evolutionary fitness, etc.) is not intended in this MS, but if/where it is, this needs to be clear.

Then there is the question of whether ASRs (regressions) need to be done. I think this would be incompatible with the data involved in the study, which is qualitative characters (i.e. phylogenetic characters; whether anatomical or habitat/general locomotor mode; mapped on a phylogeny using parsimony as the criterion for optimal mapping); not any quantitative data on locomotor function and performance. Those kinds of numerical data could be placed into regressions such as independent contrasts or PGLS, but that's not possible here as far as I know.

There is the more debatable issue of whether morphological/other non-molecular data can be used in maximum likelihood (or Bayesian) analyses, which can be used on trees with branch lengths and divergence times, and can give different results from parsimony. There is a lot in the literature about this; there are plenty of studies agreeing that ML/Bayesian analyses can be meaningfully compared with parsimony for non-molecular data (again, for building trees OR for mapping characters on them; it is more or less the same concept), but there are those that disagree. I think minimally the authors should cite references that explain this debate and justify why they are not using alternatives, unless they choose to (it doesn't need to be more than 2-3 sentences of text, if not).

I will check the revised manuscript and if it is suitable, there will not be further review. Thank you for your patience, authors, and thanks to the reviewers for their thoughtful reviews.

·

Basic reporting

No comment

Experimental design

No comment

Validity of the findings

No comment

Additional comments

The manuscript is much improved following the authors' revisions and hard work. This was always an interesting and worthy study, and now I feel the intent and implications of the findings are much clearer to the reader (particularly one who may be less familiar with Anura, sesamoids, and/or the relevant anatomy). Congratulations and well done to the authors.

I have only a few extremely minor comments which I don't feel even constitute minor revisions, and so I leave to the editor/authors' discretion on whether/how to address.

Line 58: I suggest 'muscle or tendon' instead of 'joints' leverage (as the sesamoids are thought to alter moment arm of the specific muscle/tendon passing over them)
Line 63: "the mobility of these joints is controversial" - could be helpful to have citation(s) supporting this statement?
Line 70: I suggest 'subsequently' rather than 'posteriorly', just to avoid confusion between a direction vs time sequence
Line 119: Is it explained somewhere that SM = supplementary materials?
Line 122: space missing between 8 and sp
Line 133: Tissue fixation/preservation usually significantly impacts specimen mobility. It would be interesting to know whether this is the case with the ethanol-preserved Anura (and perhaps whether the mobility of sesamoid-bearing specimens were compared with non-sesamoid-bearing specimens as a sort of control state?)
Line 135 and elsewhere: computer tomography is usually abbreviated with two capital letters (CT)
Line 141-142 and elsewhere: It might be worth changing the character labels from 1,2,3 to A,B,C to match Fig 4 and also avoid confusion (as the character states themselves are also numerically coded)
Line 196: It could be worth labelling clade 'PS' on Fig 4 for the reader's benefit?
Line 244: It's not too clear what 'the ancestral one' is referring to, maybe consider changing for clarity (e.g. the ancestral state?)
Line 254-256: Very nice explanation of these few exceptions; I suggest just adding a few words at the end of this sentence to remind the reader that these species are ones found with the palmar sesamoid (otherwise they would need to flip back through the manuscript to check sesamoid status in order to contextualise this sentence).
Fig 3A: The scale bar and arrow are quite difficult to see online and in printed version of the figure. I have found it helpful to outline the dark arrow in a light coloured border with images like this. Otherwise, a scale bar in an inset to each figure would also be nicer and easier to see.
Fig 4: Very nice, much clearer to see the correlations and exceptions. In 4C the character states label is partly missing (for 0 = terrestrial).
SM1: It is not clear what the highlighting in this file indicates

·

Basic reporting

The manuscript is well structured and the literature well cited. There are some issues in the current version of the manuscript in terms of spelling and grammar, which make some areas difficult to follow or know what the authors are trying to say.

Experimental design

The study is original, primary research which fits in the aims and scope of the journal. The research question is well defined, but there are areas where the conclusions and objectives do not match well (see comments). I was a bit confused by the methods (and do not remember being confused on the first reading), but am unsure if that is because of how they are written or my ignorance of this clade.

Validity of the findings

I have some concerns about the some of the conclusions, where are mentioned below, primarily related to the optimization conclusions and ancestral state reconstructions. As I mentioned in my first review, it would be useful to have a phylogenetically-informed regression looking at the relationship between locomotion and sesamoid presence/absence.

Additional comments

I thank the authors for taking the time to consider the previous reviews and incorporating the changes into their manuscript. Apologies for any typos or grammatical errors in this review.
I appreciate how the manuscript has been reworked to be less of an anatomical report and more of an evolutionary study.
Before the following comment, I would like to stress that I am not an expert in non-mammalian vertebrates. I found some of the Materials and Methods/Discussion difficult to follow but am unsure if that is because of how the information was presented or because of my ignorance in this area. I must leave this critique to researchers who are more knowledgeable than me in these areas.
One of the main issues I have is with some of the evolutionary language, and some of the evolutionary analyses that were being run. The author’s state in their response that they conducted an “optimization of the habitat use character” – I believe what the authors did was performed a parsimony-based ancestral-state reconstruction. If so, there are two issues here. First, this is not optimization. Optimization theory involves taking some performance metric and finding some genotype or phenotype that maximizes this performance. Often, researchers will do something like assume a positive or negative correlation between a phenotype and performance, and either use a functional morphospace or something like an OU-based evolutionary model to see how close to the (perceived) evolutionary peak the different species are. (Phylogenetically informed) correlations can be used to determine how close to the peak different species are. If I am understanding what the authors did correctly, not optimization analysis was run. If I am not, then what was done for the optimization analysis needs to be made clear in the M&M section.
Second, for the ancestral-state reconstruction, this needs to be phylogenetically-informed and not based on parsimony. My lab is currently finishing a manuscript (hopefully to be submitted in the coming month) where we did a similar type of analysis with a different clade. We looked at different evolutionary models for the sesamoids of interest, and used the best evolutionary model to perform ancestral state reconstructions. Long story short, our results were very different when we used phylogenetically-informed ASRs compared to parsimony-based ones. In our paper, we are not comparing the two as the phylogenetically-informed ASRs are more accurate, as they take factors like tree topology and branch length into account. Based on my lab’s own work, I would be worried the results of this ASR analysis are incorrect because they are based on parsimony. At a minimum, phylogenetically-informed methods should be used and compared to the parsimony-based results so we (the readers) can see if the results are similar.

As I mentioned in my first review, it would be useful to have a phylogenetically-informed regression looking at the relationship between locomotion and sesamoid presence/absence.

There are spelling and grammar mistakes throughout – I have listed some below, but it is not an exhaustive list.

Abstract
L16-17: This statement is a bit too strong. I would be surprised if, e.g., people thought that the patella was not part of the human bauplan.
L17: I would suggest saying “The palmar sesamoid in anurans is…” or something similar as the palmar sesamoid is not in all tetrapods
L19-20: “…it has been suggested that it acts by avoiding the closing of the palm, preventing grasping.” is confusing. Perhaps change to “… it has been suggested that it inhibits the closing of the palm, preventing grasping.”

L23: here and throughout – clade instead of group?

L27-8: “opening a landscape of the relationship between this element of the manus with their phylogeny” is confusing and I am not sure what the authors mean by this sentence

L31: here and throughout - CT capitalized

Results: it is not clear how these results relate to the objectives stated in the Introduction portion of the abstract, and the last sentence of the results section seems out of place, and I am unsure how this conclusion was reached given the rest of the abstract.



Introduction

L48: as above

L90-2: What did the study find? Did it support the previous sentences?

L104: toads instead of toad?

L104-5: “… can climb without any particular climbing specialization when stressful.” – I do not understand this, please clarify

L106: all of whom?


Materials and Methods

L135: Why is University of California mentioned? Morphosource should be cited as well

L138: In light of this, instead of talking about sesamoid presence/absence, why not talk about ossified sesamoid presence/absence?

L146: what version of Mesquite was used? In the abstract it says 3.5, here 2.7


Results

L154-6: The sentence is confusing and makes it sound like only four species were used in this study

L168: here and throughout – images not imagens

L187: big oval bone


Discussion

L227-9: what other factors? Selective ones? Evolutionary ones? Environmental ones?

L241: “…without crest, flat, lacks…” is confusing as written

L241-2: the discussion of crests here seems tangential

L254: I am not sure behavioral conclusions like this can be made from the data here (i.e., I do not think this conclusion can be affirmed or not with this data).

L266-8: it is not clear to me how this conclusion was reached

---

## Round 0.3 · accepted · Accept

I have checked the Tracked Changes and Rebuttal and am convinced that the revision has done a satisfactory job of addressing comments, so the manuscript is ready for publication. Congratulations!!